# Double Hyperautofluorescent Rings in Patients with USH2A-Retinopathy

**DOI:** 10.3390/genes10120956

**Published:** 2019-11-21

**Authors:** Ana Fakin, Maja Šuštar, Jelka Brecelj, Crystel Bonnet, Christine Petit, Andrej Zupan, Damjan Glavač, Martina Jarc-Vidmar, Saba Battelino, Marko Hawlina

**Affiliations:** 1Eye Hospital, University Medical Centre Ljubljana, 1000 Ljubljana, Slovenia; sustar.majchi@gmail.com (M.Š.); jelka.brecelj@kclj.si (J.B.); martina.jarcvidmar@gmail.com (M.J.-V.); marko.hawlina@gmail.com (M.H.); 2Unité de Génétique et Physiologie de l’Audition, Institut Pasteur, 75015 Paris, France; crystel.bonnet@orange.fr (C.B.); christine.petit@pasteur.fr (C.P.); 3Unité Mixte de Recherche en Santé (UMRS) 1120, Institut National de la Santé et de la Recherche Médicale (INSERM), 75015 Paris, France; 4Complexité du Vivant, Sorbonne Universités, Université Pierre et Marie Curie, Université Paris 06, 75005 Paris, France; 5Institut de l’Audition, 75012 Paris, France; 6Syndrome de Usher et Autres Atteintes Rétino-Cochléaires, Institut de la Vision, 75012 Paris, France; 7Collège de France, 75005 Paris, France; 8Department of Molecular Genetics, Institute of Pathology, Faculty of Medicine, University of Ljubljana, 1000 Ljubljana, Slovenia; andrej.zupan@mf.uni-lj.si (A.Z.); damjan.glavac@mf.uni-lj.si (D.G.); 9Department of Otorhinolaryngology and Cervicofacial Surgery, University Medical Centre Ljubljana, 1000 Ljubljana, Slovenia; saba.battelino@gmail.com; 10Faculty of Medicine, University of Ljubljana, 1000 Ljubljana, Slovenia

**Keywords:** USH2A, usher syndrome, retinitis pigmentosa, fundus autofluorescence, double hyperautofluorescent rings, electrophysiology, cone-rod dystrophy

## Abstract

*USH2A* mutation is the most common cause of retinitis pigmentosa, with or without hearing impairment. Patients most commonly exhibit hyperautofluorescent ring on fundus autofluorescence imaging (FAF) and rod-cone dystrophy on electrophysiology. A detailed study of three USH2A patients with a rare pattern of double hyperautofluorescent rings was performed. Twenty-four patients with typical single hyperautofluorescent rings were used for comparison of the ages of onset, visual fields, optical coherence tomography, electrophysiology, and audiograms. Double rings delineated the area of pericentral retinal degeneration in all cases. Two patients exhibited rod-cone dystrophy, whereas the third had a cone-rod dystrophy type of dysfunction on electrophysiology. There was minimal progression on follow-up in all three. Patients with double rings had significantly better visual acuity, cone function, and auditory performance than the single ring group. Double rings were associated with combinations of null and missense mutations, none of the latter found in the single ring patients. According to these findings, the double hyperautofluorescent rings indicate a mild subtype of *USH2A* disease, characterized by pericentral retinal degeneration, mild to moderate hearing loss, and either a rod-cone or cone-rod pattern on electrophysiology, the latter expanding the known clinical spectrum of USH2A-retinopathy.

## 1. Introduction

Mutation in *USH2A* is the most frequent cause of recessive retinitis pigmentosa (RP) as well as Usher syndrome type 2 (USH2) [1]. Regardless of the degree of hearing impartment, the development of RP is thought to be universal among patients, characterized by peripheral retinal degeneration and the predominant rod impairment on full-field electrophysiology (ERG), i.e., the “rod-cone” pattern [2,3,4,5]. We studied three *USH2A* patients with mild retinal disease, presenting with double hyperautofluorescent rings on FAF (fundus autofluorescence imaging). One patient exhibited a cone-rod pattern type of disfunction on ERG, a phenotype not yet reported in association with *USH2A*.

## 2. Materials and Methods

Three USH2A-Usher patients with hyperautoflurescent rings on FAF were recruited. Two patients (Patient 1 and 2, ages 44 and 56 years, respectively) were identified from the previously described Slovenian USH2A cohort [6] while the third patient (Patient 3, age 61 years) was identified after a targeted screen for the most frequent Slovenian *USH2A* mutation (p.Trp3955Ter) using high-resolution melt analysis (HRM) on a group of five other patients with double hyperautofluorescent rings.

The examination included Snellen visual acuity (VA), Goldmann perimetry (targets II/1 and II/4), FAF, optical coherence tomography (OCT) and microperimetry. OCT was performed using spectral-domain OCT (OCT-SLO Spectralis, Heidelberg Engineering, Dossenheim, Germany) after pupil dilation with topical 1% Tropicamide. A macular volume scan, extending 8 × 8 mm and centered at the fovea was performed in all cases. Additionally, 10 mm long linear scans were placed manually during imaging at various peripheral locations in the 360-degree circle with the aim to obtain OCT sections across the outer hyperautofluorescent rings. The external fixation light was used to guide the patient’s fixation while imaging the retinal periphery. Microperimetry (MP1, Nidek Technologies, Padua, Italy) was performed across the rings of hyperautofluorescence in one eye of Patient 1 and both eyes of Patient 3 by manually placing the targets of Goldmann target size III, 200 ms duration and 4–2 threshold strategy on the areas of interest. Electrophysiology (ERG) was performed according to the ISCEV standard [7] and all patients underwent pure tone audiometry.

A comparator group of twenty-four USH2A patients exhibiting typical single rings on FAF (12 male, average age 47 ± 13 years) was recruited from the Slovenian USH2A-Usher syndrome cohort [6]. The presence of a second peripheral hyperautofluorescent ring in the comparator group was excluded by performing peripheral FAF imaging in cases with signs of the residual peripheral visual field. The remaining patients of the Slovenian USH2A-Usher syndrome cohort had either patch or atrophy on fundus autofluorescence, thought to represent the late stages of disease following the ring [6] and were excluded.

The genetic analysis of all patients including the identification of the second mutation of Patient 3 was done using next-generation sequencing (NGS) of the *USH2A* gene as described previously [8].

Age at the exam, age of onset, visual acuity, ERG amplitudes, and the degree of hearing loss were compared between the double and single ring patients. Multiple regression analysis and the 95% confidence intervals (95% CI) of the regression lines were used for the correlation of parameters with age were used to control for the age-related disease progression.

The study was approved by Slovenian Ethics Committee for Research in Medicine and all research procedures have been carried out in accordance with The Code of Ethics of the World Medical Association (Declaration of Helsinki) for experiments involving humans. Informed consent for genetic analysis was obtained from all subjects.

## 3. Results

### 3.1. Clinical Findings in Patients with Double Hyperautofluorecent Rings

The clinical findings of patients with double hyperautofluorescent rings are summarized in Table 1. The main symptom of Patients 1 and 2 was nyctalopia that was first noted at the ages of 18 and 22 years, respectively. Patient 3 first noted visual problems at the age of 58 years, which predominantly consisted of problems involving central vision, while mild nyctalopia was only reported upon questioning. The FAF imaging revealed two hyperautofluorescent rings in 5/6 eyes and a similar incomplete pattern in one eye of the Patient 3 (Figure 1). Full-field ERG showed reduction of the cone and rod specific responses in all three patients, however, with different patterns (Figure 1 and Figure 2). The patients 1 and 2 had absent rod response and residual cone response, consistent with the diagnosis of RP, whereas the Patient 3 had notably well-preserved ERG, especially the rod response, the type of dysfunction was more consistent with cone-rod dystrophy (or near-equal rates of rod and cone loss). Goldmann perimetry in all eyes showed bilateral paracentral scotoma that corresponded to the area between the rings (Figure 1). OCT and microperimetry performed across the ring borders demonstrated that they delineate the area of impaired structure and function of the retina (Figure 3 and Figure 4). Pure tone audiometry revealed moderate hearing loss in patient 1 and mild hearing loss in patients 2 and 3 (Figure 5). Patients were followed up regularly for the next 5 years, showing minimal signs of progression (shown for Patient 3 in Figure 6).

### 3.2. Comparison with the Patients with Single Hyperautofluorescent Rings

Patients with double rings were compared with 24 single ring USH2A patients, the summary of whose clinical and genetic findings is presented in Table 2. The single ring patients had the clinical presentation typical for RP, reporting nyctalopia as their first visual symptom at the median age of 19 years (range 6–42 years) and concentrically narrowed visual fields. Those in whom electrophysiology was performed (8/24, at the median age of 27 years) had either undetectable or barely detectable light-adapted cone and dark-adapted maximal responses and undetectable dark-adapted rod responses; consistent with rod-cone dystrophy. Their hearing loss ranged from moderate to profound (Figure 5).

In comparison to the single ring patients who displayed one hyperautofluorescent ring with abnormal autofluorescence outside the ring, the double ring patients had an extra hyperautofluorescent ring on the periphery, followed by normal autofluorescence beyond that ring. The inner ring of double ring patients was comparable to the ring of the single ring patients, i.e., the OCT across the (inner) ring in single as well as double ring patients showed a transition from normal to abnormal retina with the loss of the inner segment ellipsoid (ISe) at the inner border, followed by the loss of the external limiting membrane (ELM) at the outer border of the ring (Figure 3a and Figure 4a,b; [6]). The outer hyperautofluorescent ring also corresponded to a transition between an abnormal and normal retina, however in a reverse direction. A detailed structural analysis of the outer ring was challenging as a peripheral OCT is difficult to obtain. The outer rings were accessible to imaging only in Patients 1 and 3. In these, OCT revealed a transition zone between abnormal and (relatively) normal retina. This was best demonstrated in Patient 3 whose outer ring was closest to the macula (Figure 1). The inner (macula-facing) border of the outer ring corresponded to the re-appearance of the ELM while the outer border corresponded with the re-appearance of ISe (Figure 4). In Patient 1 the transition towards better-preserved retina was also present (Figure 3b) but the layers were less discernible, possibly due to the reduced retinal thickness in the periphery. Kinetic perimetry and microperimetry in all three patients showed preserved peripheral retinal function corresponding to the structural observations.

In comparison to the single ring patients, the Patient 3 from the double ring group had significantly delayed the onset of the visual symptoms (58 years) (One-Sample Wilcoxon Signed Rank Test; *p* < 0.001) whereas Patients 1 and 2 were comparable (onset at 18 and 22 years, respectively). All double ring patients had significantly better light-adapted cone responses and dark-adapted combined rod and cone system responses (Figure 2), with the Patient 3 having the largest amplitudes. The dark-adapted rod response was detectable only in Patient 3 (close to normal) and undetectable in all other patients. Multiple regression analysis showed significant effect of age (B = −0.1; *p* < 0.01) and FAF pattern (B = −0.42; *p* < 0.01) on the VA. When plotted against age, the VA of Patient 3 was above the 95% confidence interval of the single ring group and Patients 1 and 2 were on the upper limit (Figure 5). The double ring patients had significantly better hearing than single ring patients (median 36 vs. 64 dB; Mann–Whitney U test, *p* < 0.01), and when plotted against age, Patients 2 and 3 were outside the 95% confidence interval of the single ring group (Figure 5).

### 3.3. Genetic Findings

The genotypes of patients with double rings are listed in Table 1. All had a combination of a null and a missense mutation. Both missense mutations, p.Arg303His (c.908G > A) (Patients 1 and 2) and p.Gly4032Arg (c.12094G > A) (Patient 3) are predicted to be pathogenic by Polyphen2 (genetics.bwh.harvard.edu/pph2), SIFT (https://sift.bii.a-star.edu.sg/www/SIFT_aligned_seqs_submit.html) and MutationTaster (http://www.mutationtaster.org/) and have low allele frequencies in GnomAD database (p.Arg303His present in 11/281768 alleles (0.00003904) and p.Gly4032Arg present in 1/250584 alleles (0.000003991); gnomad.broadinstitute.org, accessed 18 June 2019). Among the comparator group, 7/24 (29%) had a combination of a null and a missense mutation among which none were p.Gly4032Arg or p.Arg303His.

## 4. Discussion

The report describes the phenotypic and genetic characteristics of USH2A patients who exhibit a rare pattern of double hyperautofluorescent rings on FAF.

The multimodal analysis demonstrated that the area between the hyperautofluorescent rings corresponds with the area of photoreceptor loss. The inner and outer hyperautofluorescent rings thus represent borders between an abnormal and normal retina, but in different directions. The structure of the retina at the outer ring on OCT was analogous to that at the inner ring, showing the presence of ELM and absence of ISe, suggesting there is an absence of outer photoreceptor segments in that area [10]. The source of hyperautofluorescence, however, is difficult to derive from OCT imaging and is not fully understood. The hypotheses include increased lipofuscin accumulation in the photoreceptor inner segments and/or the RPE, possibly contributing to photoreceptor damage; or more simply, increased detection of the normal RPE autofluorescence due to decreased blockage after photoreceptor outer segment loss [11,12,13,14].

In comparison to USH2A patients exhibiting the more commonly observed single ring pattern, the patients with double rings demonstrated significantly better visual acuity and cone function, as well as significantly milder hearing loss, even when corrected for age. This suggests that the double ring pattern is associated with a relatively mild disease course. This is supported by minimal signs of disease progression in the course of five years (Figure 5). Nevertheless, the inner rings encroached the fovea in 5/6 eyes, making the prognosis regarding the central vision guarded.

Although Patients 1–3 exhibited similar FAF patterns, their phenotypes were not identical. Patients 1–2 had a relatively typical presentation of (mild) RP with nyctalopia, absent rod responses, and residual cone responses; while Patient 3 had predominantly central vision problems and surprisingly well-preserved rod responses, atypical for RP. The location of the double rings and pericentral scotoma also differed between patients; being larger and extending further into the periphery in Patients 1–2 and smaller (incomplete in one eye) in Patient 3.

Fundus autofluorescence imaging (FAF) has become one of the most important examinations for the diagnosis and management of patients with retinal dystrophies [11,15]. The most common patterns observed in RP including USH2A-Usher syndrome are hyperautofluorescent ring and central hyperautofluorescence (patch) or atrophy [6,16,17]. The hyperautofluorescent ring was well studied and was shown to represent a boundary encompassing a central retinal island of preserved structure and function [18,19,20], while hyperautofluorescent patch or atrophy were shown to be associated with longer disease duration and loss of the foveal cones [6].

The double hyperautofluorescent ring is a rarely reported FAF pattern. It was first observed in patients with *NR2E3*-dominant RP in 2012 [21,22] followed by associations with different genetic types of RP as well as other dystrophies (e.g., ABCA4-retinopathy) [23,24]; although on review the reported phenotypes may not fit exactly the same pattern in terms of the ring size, shape, and location.

There was a close spatial association between the hyperautofluorescent rings and the ring-shaped scotoma. Unlike double hyperautofluorescent rings, the ring scotoma is not an infrequent visual field pattern observed in RP patients, in some cohorts representing up to one-third of cases [25,26]. Considering these studies, it is possible that the double ring pattern is more frequent than previously thought and some clues of its clinical significance might be drawn from previous perimetry studies. Two of the latter reported similar progression in RP patients who presented with ring scotoma and those who presented with other patterns of visual loss, e.g., concentric narrowing or progressive loss from superior to inferior retina; the common endpoint of both groups being a small central island of visual field [25,26]. On the other hand, Sandberg et al. discerned two subtypes of RP depending on the location of the pericentral scotoma. They proposed that typical RP presents with an arcuate scotoma between 20–40° and reduced rod thresholds; while pericentral RP with an annular scotoma between 5–30° and near-normal rod thresholds; and have shown that the latter had slower progression [9] (Figure 1). According to this classification, Patients 1 and 2 fit more closely to the first and Patient 3 to the second subtype.

It is difficult to determine whether the two missense mutations found in our three patients are specific to the double ring phenotype. Although they have not been found in the single ring patient cohort there is a large number of different *USH2A* mutations in the population, so this is not unexpected. The mutation p.Arg303His (Patients 1 and 2) has been reported before, however without the description of FAF pattern [27,28]. Mutation p.Gly4032Arg (c.12094G > A) (Patient 3) has not been reported before and its effect is not known, however, it is possible that induces aberrant splicing due to the formation of a cryptic donor splicing site [29]. Larger cohorts with FAF data are needed to elucidate the effect of specific genotypes.

Various genes have been reported in association with the double ring pattern and/or pericentral retinal degeneration [21,22,23,24] and it is possible that the phenotype represents a milder version of retinal degeneration of different genetic backgrounds. On review, many of the reported genotypes [23,24] consist of missense mutations including all three patients in our cohort, suggesting the residual function of the affected proteins.

## 5. Conclusions

The double hyperautofluorescent rings indicate a mild subtype of USH2A disease, characterized by pericentral retinal degeneration, mild to moderate hearing loss, and either a rod-cone or cone-rod pattern on electrophysiology, the latter expanding the known clinical spectrum of USH2A-retinopathy.

## Figures and Tables

**Figure 1 genes-10-00956-f001:**
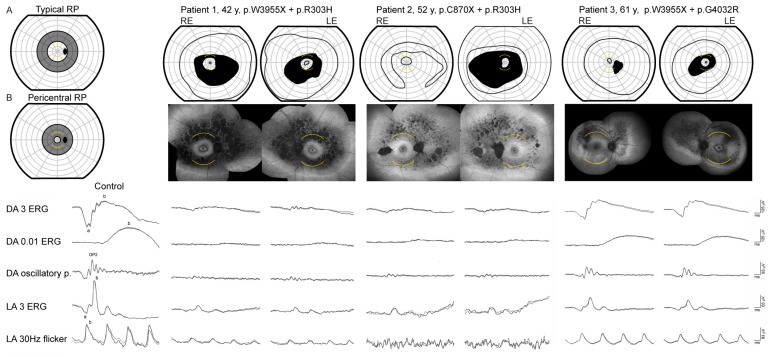
Goldmann visual field (first row) fundus autofluorescence (second row) and full-field ERG (bottom row) of three *USH2A* patients with double hyperautofluorescent rings. Goldmann perimetry was performed using II/1 and II/4 targets. The arcuate yellow lines mark the 15° radius on Goldmann visual fields and FAF images. RE = right eye, LE = left eye. The figures (**A**) and (**B**) mark the approximate location of scotoma associated with “typical” and “pericentral” RP according to Sandberg et al. [9]. Patients 1 and 2 had absent rod responses and residual cone responses in keeping with the diagnosis of RP. Patient 3 had a mild reduction of the rod (DA 0.01 ERG) and cone (LA 3 ERG and LA 30 Hz flicker) specific responses in a cone-rod dysfunction pattern. The first column shows ERG responses of a representative healthy control.

**Figure 2 genes-10-00956-f002:**
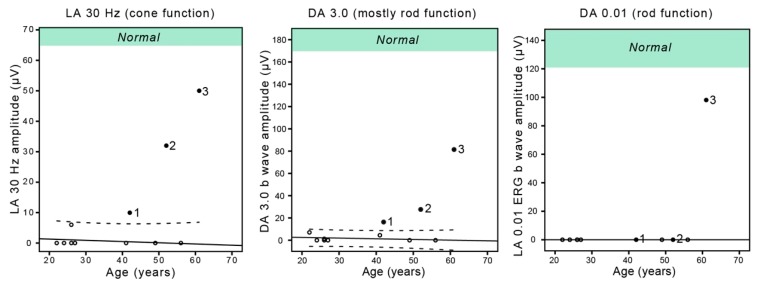
The ERG amplitudes in relation to the age of USH2A patients with double and single hyperautofluorescent rings. Filled circle = double ring, empty circle = single ring, LA = light-adapted, DA = dark-adapted. Dashed lines represent the 95% confidence intervals of the single ring group. Green area marks the normal amplitudes according to the laboratory normative. Note the good rod function in Patient 3, atypical for RP.

**Figure 3 genes-10-00956-f003:**
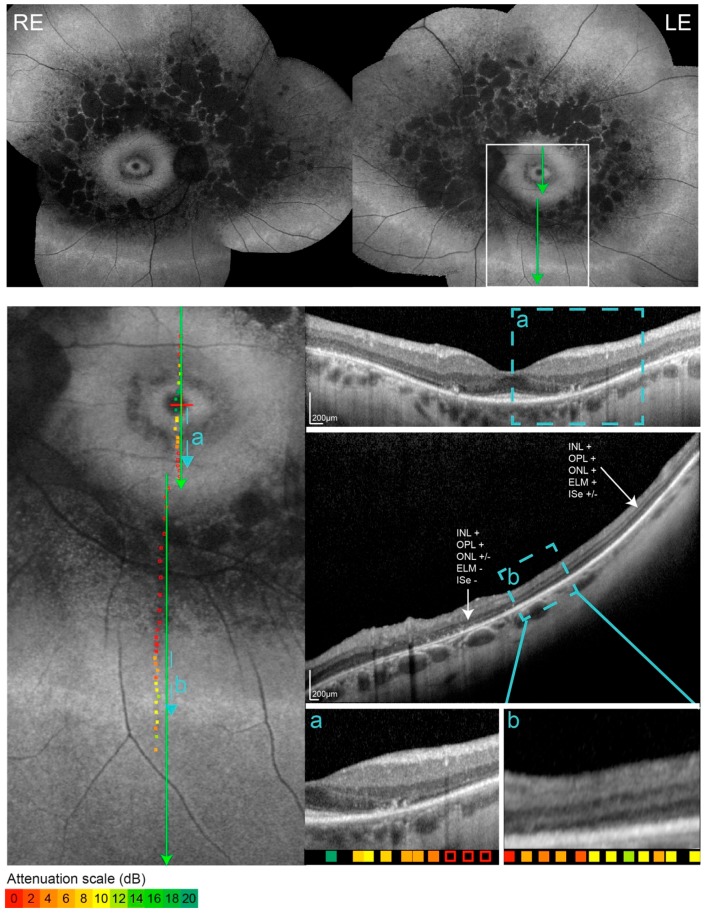
FAF, OCT and overlaid microperimetry of Patient 1. Retinal sensitivity is color-coded from green (good) to red (poor), with empty squares representing no response to the brightest stimuli. The location of each OCT scan is marked with green lines. The areas of transition trough the hyperautofluorescent borders are marked with blue lines and enlarged in rectangles **a** and **b**. The outer hyperautofluorescent ring was most accessible to the OCT imaging in the inferior retina. Panel **b** shows the re-appearance of the external limiting membrane at the location of the outer ring.

**Figure 4 genes-10-00956-f004:**
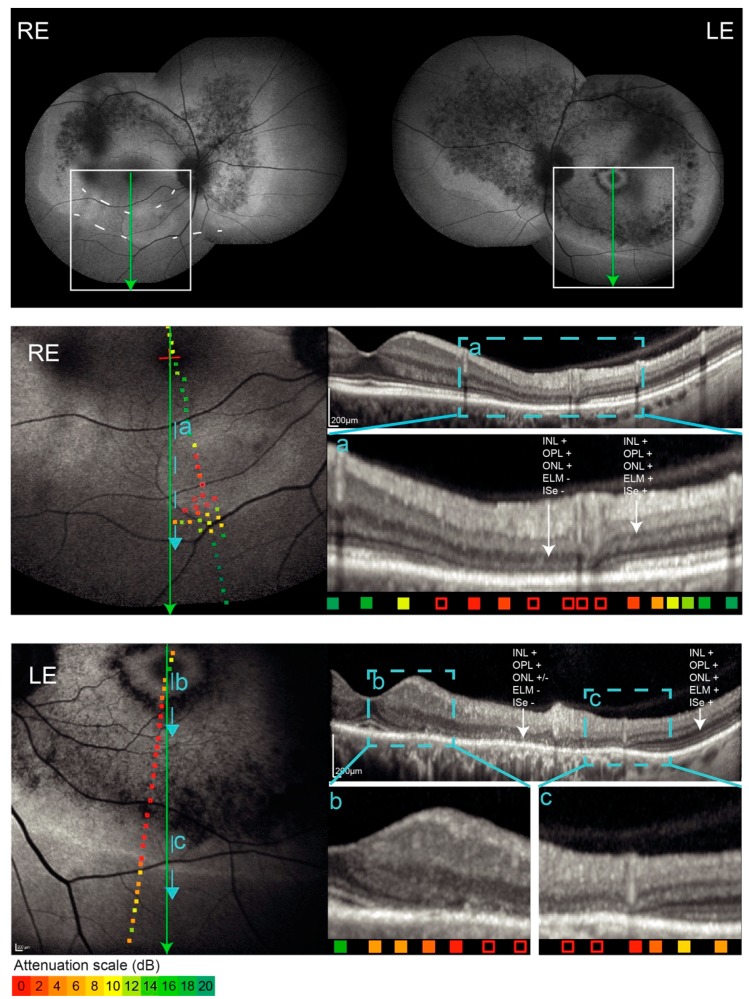
FAF, OCT, and overlaid microperimetry of Patient 3. Note the incomplete double ring pattern in the right eye. Retinal sensitivity is color-coded from green (good) to red (poor), with empty squares representing no response to the brightest stimuli. The location of each OCT scan is marked with green lines. The areas of transition trough the hyperautofluorescent borders are marked with blue lines and enlarged in rectangles a–c. The outer hyperautofluorescent ring was best accessible to OCT imaging in the inferior retina. Panel **c** shows the re-appearance of the external limiting membrane at the inner (macula-facing) border of the ring, followed by the re-appearance of inner segment ellipsoid at the outer (periphery-facing) border of the ring. Note the blood vessel on the LE FAF (dark line) as well as the panel c OCT image (hyperreflective vertical line in the inner retina), spatially corresponding with the inner border of the ring.

**Figure 5 genes-10-00956-f005:**
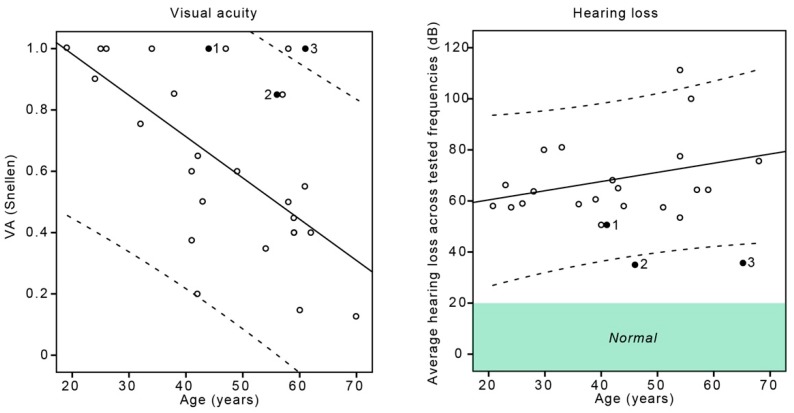
Snellen visual acuity and hearing loss in relation to the age of USH2A patients with double and single hyperautofluorescent rings. Filled circle = double ring, empty circle = single ring. Dashed lines represent the 95% confidence intervals of the single ring group. Green area marks the lower limit of normal hearing (20 dB hearing loss).

**Figure 6 genes-10-00956-f006:**
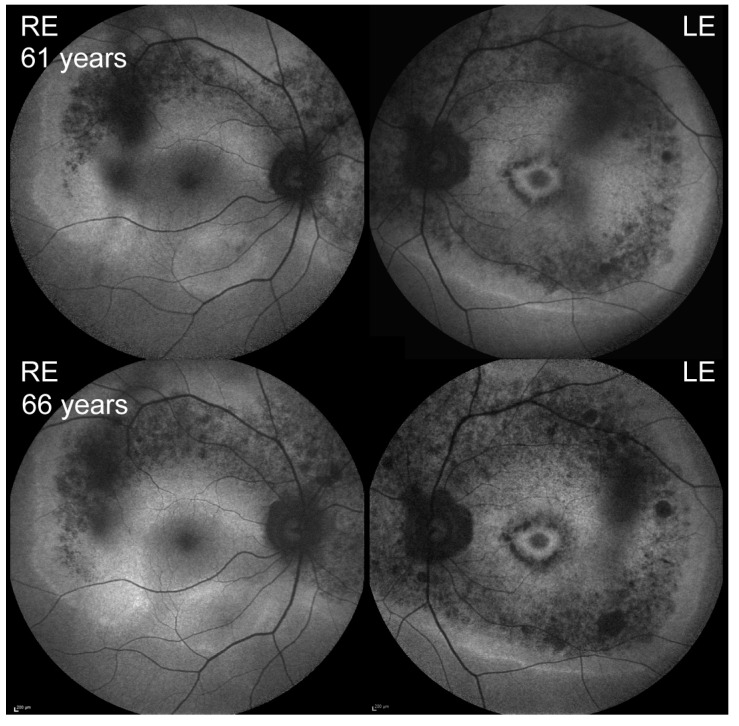
Fundus autofluorescence of Patient 3 at presentation (**top**) and after five years of follow-up (**bottom**).

**Table 1 genes-10-00956-t001:** Clinical and genetic characteristics of USH2A patients exhibiting double hyperautofluorescent rings.

ID	*USH2A* Alleles	Sex	Age (Years)	Onset (Years)	BCVA; R, L	ERG	Hearing Loss # (dB); R, L (Age)
1	p.Trp3955Ter, p.Arg303His	M	42	18	1.0, 1.0	rod-cone dystrophy	56, 45 (41)
2	p.Cys870Ter, p.Arg303His	F	52	22	0.8, 0.9	rod-cone dystrophy	34, 36 (46)
3	p.Trp3955Ter, p.Gly4032Arg	F	61	58	1.0, 1.0	cone-rod dystrophy	31, 40 (65)

BCVA = best corrected visual acuity, R = right, L = left, # = average hearing loss across the 500, 1000, 2000 and 4000 Hz (Normal = < 20 dB).

**Table 2 genes-10-00956-t002:** Summary of the clinical and genetic characteristics of the comparator group of USH2A patients exhibiting single hyperautofluorescent rings (comparator group).

N	*USH2A* Alleles	Sex	Age (Years)	Onset (Years)	BCVA	ERG	Hearing Loss # (dB)
24	75% (18/24) null + null; 25% (6/24) null + missense	12 M, 12 F	median 45, range 19–70	median 19, range 6–42	median 0.6, range 0.1–1.0	rod-cone dystrophy in 8/8 performed, at the median age of 27 y	Median 64 dB, range 51–111 dB) in 21 performed, at the median age of 42 y *

N = number of patients. M = male F = female BCVA = best corrected visual acuity # = average hearing loss across the 500, 1000, 2000, and 4000 Hz (Normal = < 20 dB). * The audiogram was performed in 21 patients however all 24 patients reported congenital hearing loss. The average values between the right and left eye or ear were used.

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
