# Peer review of "Double Hyperautofluorescent Rings in Patients with USH2A-Retinopathy"

_genes, 2019, doi:10.3390/genes10120956_

Round 1

Reviewer 1 Report

The paper is original, well-written and well-organized.

Three is a low number but absolutely enough for such a rare pathology characterized by even more rare pattern of double hyperautofluorescent rings.

Methods are undoubtedly very detailed. Authors specified each test/examination taken on single patient. Definitely good-quality images. Diagrams display large amounts of information in ways that are easy to understand. Updated, essential and redundant bibliography.

In Table 1, please correct n. 3 “rone-rod” in “cone-rod”.

Author Response

N. 3 “rone-rod” in Table 1, was corrected to “cone-rod”.

Reviewer 2 Report

This study analyzed the clinical presentation of 3 retinitis pigmentosa patients in USH2A mutation, with a rare pattern of double hyperautofluorescent rings in the fundus. The authors found that patients with double rings had significantly better visual acuity, cone function and auditory performance than classic single ring group. The authors concluded that double hyperautofluorescent rings indicate a mild subtype of USH2A disease.

Page 2, Line 50. Please describe the detailed methods of the OCT acquisitions. Did you perform OCT en-face images?

Figure 3 and Figure 4. How may scans of OCT did you perform on the retina? The scans should be performed in different direction and get the similar results in the 360 degree circle of the rings.

Please present the detailed high magnified OCT images to compare the details between the hyper- and hypo- autofluorescent region in double hyperautofluorescent rings. Also, compare the difference between single and double autofluorescent rings.

Based on the findings of the OCT and FAF, please discuss the origin of this extra hyperautofluorescent rings histologically.

Author Response

Response to the reviewer 2 

We thank the reviewer for kind and insightful  comments.

1) Page 2, Line 50. Please describe the detailed methods of the OCT acquisitions. Did you perform OCT en-face images?

The following text was added to describe the methods of the OCT acquisitions: "OCT was performed using spectral-domain OCT (OCT-SLO Spectralis, Heidelberg Engineering, Germany) after pupil dilation with topical 1% Tropicamide. A macular volume scan, extending 8 x 8 mm and centered at the fovea was performed in all cases. Additionally, 10 mm long linear scans were placedmanually during imaging at various peripheral locations in the 360 degree circle with the aim to obtain OCT sections across the outer hyperfluorescent rings. The external fixation light was used to guide patient's fixation while imaging the retinal periphery."

En-face imaging was not performed.

2) Figure 3 and Figure 4. How may scans of OCT did you perform on the retina? The scans should be performed in different direction and get the similar results in the 360 degree circle of the rings.

This has been now been explained in Methods - see above.

The following sentence was added to the figure 3: "The outer hyperautofluorescent ring was best accessible to the OCT imaging in the inferior retina. Panel b shows the re-appearance of external limiting membrane at the location of the outer ring."; and to the figure 4: "The outer hyperautofluorescent ring was best accessible to OCT imaging in the inferior retina. Panel c shows the re-appearance of external limiting membrane at the inner (macula-facing) border of the ring, followed by the re-appearance of inner segment ellipsoid at the outer (periphery-facing) border of the ring. Note the blood vessel on the LE FAF (dark line) as well as the panel c OCT image (hyperreflective vertical line in the inner retina), spatially corresponding with the inner border of the ring."

3.1.) Please present the detailed high magnified OCT images to compare the details between the hyper- and hypo- autofluorescent region in double hyperautofluorescent rings.

The figures have been carefully selected to show the pathology in the most representative way and are of the highest possible quality provided by the current equipment (OCT-SLO Spectralis, Heidelberg Engineering, Germany). Although OCT images in the peripheral retina are difficult to obtain and higher magnification is not possible, we took the utmost care that the information was not lost by zooming in and additional description of the features was added .

3.2.) Also, compare the difference between single and double autofluorescent rings.

The following was added in Results: "In comparison to the single ring patients who displayed one hyperautofluorescent ring with abnormal aurofluorescence outside the ring, the double ring patients had an extra hyperautofluorescent ring on the periphery, followed by normal autofluorescence beyond that ring. The inner ring of double ring patients was comparable to the ring of the single ring patients, i.e. the OCT across the (inner) ring in single as well as double ring patients showed a transition from normal to abnormal retina with the loss of the inner segment ellipsoid (ISe) at the inner border, followed by the loss of the external limiting membrane (ELM) at the outer border of the ring (Fig. 3.a and Fig. 4 a,b; [6]).

The outer hyperautofluorescent ring also corresponded to a transition between abnormal and normal retina, however in a reverse direction. A detailed structural analysis of the outer ring was challenging as a peripheral OCT is difficult to obtain. The outer rings were accessible to imaging only in patients 1 and 3. In these, OCT revealed a transition zone between abnormal and (relatively) normal retina. This was best demonstrated in the patient 3 whose outer ring was closest to the macula (Fig. 1). The inner (macula-facing) border of the outer ring corresponded to the re-appearance of the ELM while the outer border corresponded with the re-appearance of ISe (Fig. 4). In the patient 1 the transition towards better preserved retina was also present (Fig. 3.b) but the layers were less discernible, possibly due to the reduced retinal thickness in the periphery. Kinetic perimetry and microperimetry in all three patients showed preserved peripheral retinal function corresponding to the structural observations."

4) Based on the findings of the OCT and FAF, please discuss the origin of this extra hyperautofluorescent rings histologically.

The following was added in Discussion: "The inner and outer hyperautofluorescent rings thus represent borders between abnormal and normal retina, but in different directions. The structure of the retina at the outer ring on OCT was analogous to that at the inner ring, showing the presence of ELM and absence of ISe, suggesting there is an absence of outer photoreceptor segments in that area [20]. The source of hyperautofluorescence however is difficult to derive from OCT imaging and is not fully understood. The hypotheses include increased lipofuscin accumulation in the photoreceptor inner segments and/or the RPE,  possibly contributing to photoreceptor damage; or more simply, increased detection of the normal RPE autofluorescence due to decreased blockage after photoreceptor outer segment loss [9, 21-23]."

Please see attachment for the revised version of the paper.

Round 2

Reviewer 2 Report

Thanks. I have no more comments.